# Different Re-Irradiation Techniques after Breast-Conserving Surgery for Recurrent or New Primary Breast Cancer

Camille Hardy Abeloos [1], Juhi M. Purswani [1], Paulina Galavis [1], Allison McCarthy [1], Christine Hitchen [1], J. Isabelle Choi [2] and Naamit K. Gerber [1,*]

[1]    Department of Radiation Oncology, NYU School of Medicine, New York, NY 10016, USA
[2]    Department of Radiation Oncology, Memorial Sloan Kettering Cancer Center, New York, NY 10065, USA
*    Correspondence: naamit.gerber@nyulangone.org

**Abstract:** Breast re-irradiation (reRT) after breast-conserving surgery (BCS) using external beam radiation is an increasingly used salvage approach for women presenting with recurrent or new primary breast cancer. However, radiation technique, dose and fractionation as well as eligibility criteria differ between studies. There is also limited data on efficacy and safety of external beam hypofractionation and accelerated partial-breast irradiation (APBI) regimens. This paper reviews existing retrospective and prospective data for breast reRT after BCS, APBI reRT outcomes and delivery at our institution and the need for a randomized controlled trial using shorter courses of radiation to better define patient selection for different reRT fractionation regimens.

**Keywords:** breast re-irradiation; toxicity; outcome; eligibility

## 1. Introduction

The optimal treatment of isolated ipsilateral breast tumor recurrence (IBTR) is not well defined. Given higher in-breast tumor recurrence (IBTR) rates following breast-conserving surgery (BCS) alone and concern for increased adverse events with a second course of radiation, mastectomy remains the standard of care [1–5]. Adverse events are reported and graded using the Common Terminology Criteria for Adverse Events (CTCAE) scale detailed in Table 1. These can be further subdivided into acute adverse events defined as ≤3 months since completion of radiation or late adverse events defined as >3 months since completion of radiation. Breast cosmesis is also often reported by using the scoring system derived by Harris et al.: excellent, good and fair [6].

**Table 1.** Common terminology criteria for adverse events grade definitions.

| Grade | Definition |
|---|---|
| 1 | Mild; asymptomatic or mild symptoms; clinical or diagnostic observations only; intervention not indicated |
| 2 | Moderate; minimal, local or noninvasive intervention indicated; limiting age-appropriate instrumental activities of daily living |
| 3 | Severe or medically significant but not immediately life-threatening; hospitalization or prolongation of hospitalization indicated; disabling; limiting self-care activities of daily living |
| 4 | Life-threatening consequences; urgent intervention indicated |
| 5 | Death related to adverse event |

With improved radiation techniques, recent publications have shown lower rates of toxicity and similar outcomes with the addition of partial-breast re-irradiation after BCS compared to salvage mastectomy (SM) [7–9].

This paper reviews available and ongoing studies addressing patient selection, radiation delivery and outcomes in patients presenting with a recurrent or new primary breast cancer. This paper also presents accelerated partial-breast irradiation (APBI) delivery and dose constraints used at our institution for re-irradiation.

## 2. Studies Comparing Salvage Mastectomy with Repeat Breast-Conserving Surgery

There are no randomized phase III trials comparing SM and BCS followed by reRT for IBTR initially treated with breast-conserving surgery and radiation. A matched-pair analysis by the GEC-ESTRO Breast Cancer Working Group compared SM (377 patients) and BCS followed by reRT using interstitial multicatheter brachytherapy (377 patients) [7]. With a median follow-up of 75.4 months, there were no significant differences between SM and conservative treatment in 5-year overall survival (OS) (88% vs. 86%, respectively) and cumulative incidence of third breast events (2.3% vs. 2.8%, respectively). Similarly, a recent SEER analysis of 3648 patients between 1999 and 2015 with IBTR ≤2 cm showed no statistically significant differences in OS and breast cancer specific survival (BCSS) in 2831 patients receiving mastectomy compared to 817 patients receiving BCS followed by radiation [10]. BCSS was defined as the time from the date of surgery for IBTR to the date of death from breast cancer. OS was defined as the time from the date of surgery for IBTR to the date of death or last follow-up. With a median follow-up of 45 months, the multivariate Cox model showed that the omission of radiation was associated with worse OS and BCSS. This established the role of breast reRT after repeat BCS. Both studies therefore showed that repeat BCS and reRT was an effective alternative to salvage mastectomy in women presenting with a first recurrence after an initial breast conservation approach.

## 3. Re-Irradiation Following Repeat Breast-Conserving Surgery

### 3.1. Retrospective Studies Using Brachytherapy-Based Techniques and Intraoperative Radiation Therapy

Historically, the most used partial-breast re-irradiation technique was multi-catheter interstitial brachytherapy (MIB). For this technique, catheters are placed in the breast around the lumpectomy cavity. Experience and training are required to place these catheters and determine the optimal dose distribution. Doses of radiation can be delivered directly to the tumor bed via these catheters using a high dose rate, low dose rate or pulsed dose rate. The largest study to date using MIB was conducted by the GEC-ESTRO breast cancer group with a total of 217 patients [8]. At a median follow-up of 3.9 years, the GEC-ESTRO study reported a second local recurrence rate of 7% at 10 years. These findings were consistent with smaller institutional series [11–18]. In regard to late adverse events, they reported 50% grade 1, 39% grade 2, 10% grade 3 and 1% grade 4 events. The most common types of late radiation effects were cutaneous and sub-cutaneous fibrosis (67%), telangiectasia (16%) and hyperpigmentation (9%). Among the 52% of women who underwent cosmetic assessment, 85% of them received an excellent or good cosmetic score.

Intraoperative radiation therapy (IORT) has also emerged as an alternate re-irradiation technique with the benefit of performing salvage lumpectomy and reRT at the same time in a single procedure [19–22]. It may also be more user friendly than brachytherapy, which requires more procedural technique and training. Elfgen et al. showed no significant difference in disease-free survival in 26 patients treated with BCS and IORT vs. 35 patients treated with mastectomy alone vs. 52 patients treated with mastectomy with reconstruction [23]. IORT was delivered using IntraBeam system with 20 Gy prescribed to applicator surface. The Intrabeam system is a 50 kV photon beam mobile X-ray unit. IORT is delivered using spherical applicators ranging from 2 to 5 cm in diameter and treatment time varies from 20 to 45 min depending on the applicator size used. Boehm et al. performed the largest study to date with a total of 57 patients also treated with the Intrabeam system with 20 Gy prescribed to applicator surface [22]. At a median follow-up of 24.3 months, the locoregional control rate was 89%. With a longer median follow-up of 58 months, Thangarajah reported a locoregional control rate of 90% in 41 patients treated similarly with

50 KV photons to 20 Gy [19]. There are different IORT technologies, including the Axxent system, which is an electronic low-energy X-ray tube integrated into a flexible multi-lumen catheter producing 40–50 Kv X-rays at the catheter tip. Mobile electron accelerators can also be used to deliver electron radiation with a cone inserted intraoperatively. Blandino et al. reported outcomes in 29 patients treated with IORT with electron beam delivered by a mobile linear accelerator [20]. Patients received a single dose of 18 Gy prescribed to 90% isodose. Radioprotection of the thoracic wall was achieved using steel shielding disks placed in the lumpectomy cavity. Since electrons are more penetrating than low-energy photons, shielding is required to protect the underlying tissue. Blandino et al. reported a 5-year local control rate of 92.3% using IORT with electron beam, comparable to the local control rate of 90% reported by Thangarajah using low-energy photons [19].

### 3.2. Retrospective Studies Using External Beam Radiation Therapy

Given its accessibility and ease of use, external beam radiation therapy (EBRT) is increasingly being used following second BCS. It is safe and effective with low rates of toxicity and in-breast tumor recurrence rates equivalent to those achieved with brachytherapy or IORT, ranging from 0 to 10%, as shown in Table 2 [24–30].

The first two studies evaluated the use of external beam radiation with electrons to the operative quadrant of the breast to a dose of 5000 cGy in 25 fractions [24,25]. Mullen et al. reported a lower rate of third IBTR with a longer median follow-up (12% vs. 21% reported by Deutsch et al. [24,25]. However, the study by Mullen et al. included only 17 total patients compared to 39 patients included in the study by Deutsch et al. Janssen et al. evaluated the use of external beam radiation with photons to a dose of 45 Gy (1.8 Gy daily) in 83 patients [26]. Half of the patients received radiation to the partial breast and half received radiation to the mastectomy scar. At median follow-up of 35 months (range: 3–143 months), 12 patients had a local recurrence (14%) with no grade 3 or worse toxicity [26]. With a different fractionation regimen of 45 Gy in 1.5 Gy twice daily in a cohort of 34 patients, Chen et al. reported 2 patients with locoregional recurrence (5.9%) at a median follow-up of 23.5 months [27]. Similarly, there was no grade 3 or higher toxicity. Of note, all patients were treated to the partial breast. Despite variations in technique and dose, these studies showed low in-breast tumor recurrence rates and low rates of toxicity.

More recently, proton therapy is under investigation. The primary rationale for using protons is their potential to decrease toxicity by decreasing the dose delivered to normal tissue. In the Proton Collaborative Group prospective multicenter registry analysis of 50 patients receiving proton beam reRT to a median of 55.1 RBE Gy, only 3 patients had a local recurrence at a median follow-up of 12.7 months (range: 0–41.8 months)(28). Of note, most of these patients received chest wall radiation with or without regional nodes (70%). Of the 50 patients, 52% received uniform scanning, 24% received pencil beam scanning and 24% were not reported. Acute grade 3 toxicities were experienced by 5 patients (10%), including pain ($n = 2$, 4%), radiation dermatitis ($n = 1$, 2%), lymphedema ($n = 1$, 2%), and anorexia ($n = 1$, 2%). Late grade 3 toxicities were experienced by 4 patients (8%), including pain ($n = 2$, 4%), radiation dermatitis ($n = 2$, 4%), and wound infection ($n = 1$, 2%). In an analysis of the Memorial Sloan Kettering Cancer Center institutional experience, at a median follow-up of 21 months, Choi et al. reported no local or regional recurrence in 46 patients receiving uniform (70%) or pencil beam scanning (30%) proton RT [29]. Similar to the multicenter registry analysis, most of these patients underwent radiation to chest wall +/− RNI (71.6%). Acute grade 3 toxicities were experienced by 14 patients (30%), all of which were acute grade 3 dermatitis, and 1 patient experienced late grade 3 breast pain. On backwards selection multivariable analysis, no dosimetric parameters were found to be significant predictors of grade $\geq$ 2 acute or late toxicities. Nine of thirteen (69.2%) patients who underwent implant or flap reconstruction developed capsular contracture, three (23.1%) requiring surgical intervention. Similarly, LaRiviere et al. reported outcomes in 27 patients receiving proton reRT to CW +/− RNI [30]. At a median follow-up of 16.6 months, 1 patient developed an in-field local recurrence, 2 patients experienced acute

grade 3 dermatitis, 2 patients had acute grade 3 breast pain, 1 patient had late grade 3 dermatitis and 1 patient had late grade 4 dermatitis and 1 patient had late grade 3 breast pain (30). All three studies using protons showed equivalent IBTR rates to those obtained with photon based treatment. Of note, proton treatment is more frequently delivered to the chest wall compared to photon treatment with slightly higher rates of dermatitis.

*3.3. Prospective Trials Using Brachytherapy-Based Techniques and External Beam Radiation Therapy*

There are currently few prospective trials reporting efficacy and safety outcomes of breast reRT, as shown in Table 3 [31–36].

The University Hospitals Cleveland Medical Center in Ohio led the first prospective phase II trial evaluating implant radiation therapy with either balloon brachytherapy (34 Gy/10 fractions over 5 days) or IORT (50 kv, single fraction) for breast reRT [31]. From 2008 to 2014, 13 patients were enrolled. In the first patient undergoing balloon brachytherapy, there was difficulty with tissue-balloon conformation resulting from prior radiation, so the remaining 12 patients were only offered IORT. At a median follow-up of 7.8 years, one patient in the IORT group had local recurrence at 2 years after her second course of radiation yielding a local recurrence rate of 8%. At baseline, before reRT, 80% of patients reported excellent–good cosmetic outcomes, compared with 42% at their 5-year postop visit. From baseline to 5 years post-treatment, there was the largest increase in skin atrophy and hyperpigmentation and smallest change in erythema and skin telangiectasia, as documented by physician assessment.

RTOG 1014 was the first prospective phase II trial evaluating salvage lumpectomy and adjuvant 3D conformal external beam partial-breast reRT with conventional fractionation [32]. Eligibility criteria included a unifocal breast recurrence ≤3 cm occurring more than 1 year after breast-conserving therapy and excised with negative margins. Between June 2010 and June 2013, a total of 65 women were enrolled and treated with repeat BCS followed by 45Gy delivered in 1.5Gy fractions BID using external beam 3D-CRT. At a median follow-up of 5.5 years, four patients had breast cancer recurrence (5%). Both distant metastasis-free survival and overall survival rates were 95%. There were only 4 patients (7%) who had late grade 3 toxicity events with no grade 4 or 5 events. The majority of grade 3 events were breast fibrosis. Both RTOG 1014 and phase II trial led by the University Hospitals Cleveland Medical Center only included early stage unifocal local recurrent invasive or non-invasive carcinoma excluding lobular carcinoma in situ and tumor size ≤3 cm. Both showed the safety and feasibility of breast reRT with 3D CRT in these early stage patients. In RTOG 1014, patients with 0 to 3 positive axillary lymph nodes without extracapsular extension were also eligible for enrollment, although all patients were clinically node-negative and those who underwent nodal sampling were pathologically confirmed node-negative (25%).

A prospective phase II trial is ongoing in France to further evaluate the feasibility of repeated breast-conserving surgery combined with reRT using IORT 20 Gy/1 fraction (NCT02386371) [33]. In the primary setting, two published randomized trials have failed to demonstrate non-inferiority between IORT and whole-breast irradiation [37,38]. Given the lack of consistent data, this technique should be used cautiously in patients after repeat BCS. There is also an ongoing phase II trial led by the New York Proton Center evaluating proton external beam partial-breast reRT to 40 Gy RBE in 10 daily fractions [34]. Finally, there is an ongoing prospective phase II trial evaluating the use of APBI with the target volume encompassing the entire surgical bed with 1.0–1.5 cm margins to a dose of 30 Gy in 5 fractions daily [36].

**Table 2.** Retrospective studies using external beam radiation therapy corresponds to Section 3.2 in text.

| Trial | Years of Enrollment | N/FU | Median Time between RT Courses (yr) | Technique | RT Target | ReRT Dose | IBTR % (nb of Patients) | Toxicity (nb of Patients) |
|---|---|---|---|---|---|---|---|---|
| Mullen E et al. [24] University of Pittsburgh | 1997 | N = 17 75 months | 2.5 (0.83–10.8) | electron | partial breast (operative quadrant) | 50 Gy in 2 Gy daily | 12% (2) | unknown |
| Deutch M et al. [25] University of Pittsburgh | 2002 | N = 39 51.5 months | 5.25 (1.3–24.2) | electron | partial-breast (operative quadrant) | 50 Gy in 2 Gy daily | 20.5% (8) | excellent cosmesis (12) good cosmesis (15) poor cosmesis (9) |
| Janssen S et al. [26] University of Luebeck, Germany | March 2004–October 2016 | N = 83 35 months | 9.75 (1.3–29.7) | 3D CRT | partial-breast (51%) or mastectomy scar (49%) | 45 Gy in 1.8 Gy BID | 14% (12) | no grade 3 or higher |
| Chen at al. [27] Memorial Sloan Kettering | 2011–2019 | N = 34 23.5 months | 9.8 (1.4–27.2) | 3D-CRT | partial-breast | 45 Gy in 1.5 Gy BID | 6% (2) | no grade 3 or higher |
| Proton Collaborative Group (PCG) multicenter analysis [28] | 2011–2016 | N = 50 12.7 months | 8.65 (0.45–35.9) | proton | chest wall +/- RNI (70%), RNI (20%), whole-breast +/− RNI (8%), partial-breast (2%) | 55.1 RBE Gy (45.1–76.3) | 6% (3) | acute grade 3: pain (2), dermatitis (1), lymphedema (1), anorexia (1) late grade 3: pain (2) dermatitis (2), wound infection (1) |
| Choi et al. Memorial Sloan Kettering [29] | January 2012–August 2020 | N = 46 21 months | 7 (0.42–30) | proton | CW +/− RNI (78.1%), RNI (10.9%), whole-breast (2.2%), partial-breast (8.7%) | 50.4 RBE Gy (40–66.6 Gy) | 0% (0) | acute grade 3: dermatitis (14) late grade 3: breast pain (1) |
| LaRiviere et al. University of Pennsylvania [30] | 2012–2019 | N = 27 16.6 months | 9.7 (0.9–37.6) | proton | CW + regional nodes (81.5%) or CW alone (11%) or regional nodes alone (7.5%) | median dose: 51 Gy (49.5–51) in 1.5 Gy BID | 1 (4%) | acute grade 3: dermatitis (2 patients), breast pain (2 patients) late grade 3: breast pain (1), dermatitis (1) late grade 4: dermatitis (1) |

Abbreviations: in-breast tumor recurrence (IBTR), radiation (RT), re-irradiation (ReRT), 3D conformal radiation therapy (3D-CRT), regional nodal irradiation (RNI), chest wall (CW), breast-conserving surgery (BCS), salvage mastectomy (SM), accelerated partial-breast irradiation (APBI), intraoperative radiation therapy (IORT).

**Table 3.** Prospective breast re-irradiation trials (corresponds to sect ion 3.3 in text).

| Trial | Years of Enrollment | N/FU | Eligibility | Technique | Dose | IBTR % (nb of Patients) | Toxicity |
|---|---|---|---|---|---|---|---|
| Case Comprehensive Cancer Center (NCT00945061) [31] | September 2008–July 2018 | N = 13 7.8 yrs | age ≥ 18 tumor size ≤ 3 cm cN0 negative surgical margins (>2 mm or negative re-excision) chemotherapy > 2 weeks after RT | electron IORT or interstitial brachytherapy | electron IORT: 21 Gy/1 fx brachytherapy: 34 Gy/10 fx (3.4 Gy BID) | 7.5% (1 patient in IORT group) | increase in skin atrophy, hyperpigmentation, fibrosis, skin induration in pretreatment vs. 4 yr post-treatment |
| RTOG 1014 (NCT01082211) [32] | June 2010–June 2013 | N = 65 5.5 yrs | age ≥ 18 recurrence > 1 yr after initial BCS + RT tumor size ≤ 3 cm negative surgical margins (no tumor on ink) 0–3 positive axillary LN without ECE | 3D-CRT | 45 Gy/30 fx (1.5 Gy BID) | 6% (4) | 4 patients had grade 3; no grade 4 or 5 |
| Institut du Cancer de Montpellier-Val d'Aurelle (NCT02386371) [33] | March 2014–June 2020 | active, not recruiting | age ≥ 50 years recurrence > 5 yr after initial BCS + RT tumor size ≤ 2 cm cN0 | IORT | IORT: 20 Gy/1 fx | | |
| New York Proton Center (NCT01766297) [34] | February 2013–January 2025 | recruiting | age ≥ 50 years recurrence > 1 yr after initial BCS + RT tumor size ≤ 3 cm negative surgical margins (no tumor on ink) cN0 | protons | 40 Gy (RBE)/10 fx | | |
| PD7-09 Espinosa-Bravo M Barcelona, Spain [35] | 2014–2020 | completed | age ≥ 50 years old >4 yr from primary treatment tumor size ≤ 2 cm no history of major RT toxicity | photons | 40.05 Gy (2.67 Gy/fx) | 2.85 % (1 patient) | Not reported |
| Personalized Second Chance Breast Conservation (PSCBC) (NCT04371913) [36] | June 2020–December 2022 | recruiting | age ≥ 18 recurrence > 1 yr after initial BCS + RT tumor size ≤ 3 cm negative surgical margins (no tumor on ink) 0–3 positive axillary LN without ECE | 3D-CRT | 30 Gy/5 fx | | |

Abbreviations: in-breast tumor recurrence (IBTR), intraoperative radiation treatment (IORT), 3D conformal radiation therapy (3D-CRT).

These ongoing prospective trials all include unifocal early stage invasive or non-invasive breast carcinoma recurrence. There were a few key differences in eligibility criteria between U.S. and European based trials: tumor size ≤3 cm in trials based in the United States and tumor size ≤ 2 cm in trials based in Europe. European trials include patients with a longer time interval between recurrence and initial BCS and RT: >1 year in U.S. trials and >4–5 years in European based trials. Some trials also only included patients ≥ 50 years old while others included patients ≥ 18 years old.

### 3.4. Re-Irradiation with External Beam Radiation Therapy with Hypofractionation/Ultrahypofractionation

Given the modest rates of observed toxicity with 3D-CRT, there is increased interest in using hypofractionated (40 Gy in 15 fractions daily) and ultrahypofractionated (30 Gy in 5 fractions every other day) regimens in the setting of reRT. The hypofractionated regimen of 40 Gy in 15 fractions daily is often more feasible than the RTOG 1014 regimen delivered twice daily. The ultrahypofractionated course of 30 Gy in 5 fractions every other day also shortens overall treatment time. In the primary setting, the IMPORT LOW trial randomized over 2000 patients to 40 Gy whole-breast radiotherapy (control), 36 Gy whole-breast radiotherapy and 40 Gy to the partial breast (reduced-dose group), or 40 Gy to the partial breast only (partial-breast group) in 15 daily treatment fractions [39]. It showed no difference in IBTR between all three groups. Patients further reported on cosmetic outcomes by completing EORTC questionnaires and protocol specific questions assessing overall skin appearance change, breast appearance change, breast hardness or firmness to touch and shoulder stiffness. At the 5-year assessment, patients reported fewer moderate or marked events of breast becoming harder or firmer in both the reduced-dose group and partial-breast group compared with the whole-breast group. Of note, the IMPORT LOW trial was not an accelerated regimen as the fractionation was identical in the whole-breast and partial-breast arms. Using an accelerated partial-breast regimen of 30 Gy in five 6 Gy fractions, the Florence trial showed improved acute and late adverse events and excellent to good cosmetic outcomes in 90% of patients treated with accelerated PBI vs. WBI [40]. Using the fractionation regimen in patient treated in the prone position, Formenti et al. reported a similar patient-rated cosmetic outcome with 89% of patients rating their cosmesis as good to excellent at a median follow-up of 64 months [41].

Our institution conducted a retrospective review of 66 patients from two institutions receiving different breast reRT fractionations including 38 patients treated with 45 Gy (1.5 Gy BID), 18 patients treated with 45 Gy (1.8 Gy daily), 4 patients treated with hypofractionation (2.67 Gy daily), 3 patients treated with ultrahypofractionation (6 Gy every other day) and 3 patients treated with whole-breast conventional fractionation (1.8–2 Gy daily). At a median follow-up of 16.5 months (range: 3–60 months), there was only one regional recurrence in a patient's right axilla 10 months after completing partial-breast reRT 45 Gy in 1.5 Gy BID. The patient had a short interval between her two radiation courses of 2 years with her initial course consisting of hypofractionated WBRT to 42.4 Gy in 16 fractions with 10 Gy boost. Out of the 3 patients who received ultrahypofractionated partial-breast reRT, 2 patients failed outside the breast, 1 with a metastatic pleural effusion 14 months after her second course of radiation and the second patient with a contralateral breast IDC 26 months after completion of her second course of radiation. Out of the 4 patients who received hypofractionated partial-breast reRT, none presented with regional recurrence or distant failure.

Our study showed that partial-breast re-irradiation could be delivered safely with minimal toxicity. One patient experienced acute grade 3 dermatitis after undergoing right PBI with mammosite balloon brachytherapy to 34 Gy in 1 fraction during her first course of radiation and WBRT to 46.8 Gy in 26 daily fractions with 14 Gy lumpectomy cavity boost in 7 fractions during her second course of radiation. The interval time between her first and second course was 12 years. She experienced significant dermatitis of the whole breast during her second course of radiation. She subsequently developed late grade

3 telangiectasia at 1- and 2-year follow-up in the right upper outer quadrant where her mammosite balloon was previously inserted and in the right upper inner quadrant where she had the briskest dermatitis from her second course, illustrated in Figure 1. One patient developed grade 3 fibrosis 3 years after undergoing WBRT to 50 Gy in 25 fractions with a 10 Gy lumpectomy cavity boost in 5 fractions and partial-breast reRT 45 Gy in 30 fractions twice daily to her recurrent disease with an 18 year interval between courses. There were no other late grade 3 events and no grade 4 or 5 events. Out of the 3 patients who received ultrahypofractionated partial-breast reRT, 2 patients experienced late grade 2 fibrosis. Out of the 4 patients who received hypofractionated partial-breast reRT, 1 patient experienced late grade 2 fibrosis and 1 patient experienced late grade 2 breast atrophy. No fractionation regimen was found to be a significant predictor of ≥ grade 2 events on univariate analysis though it was a small cohort of patients.

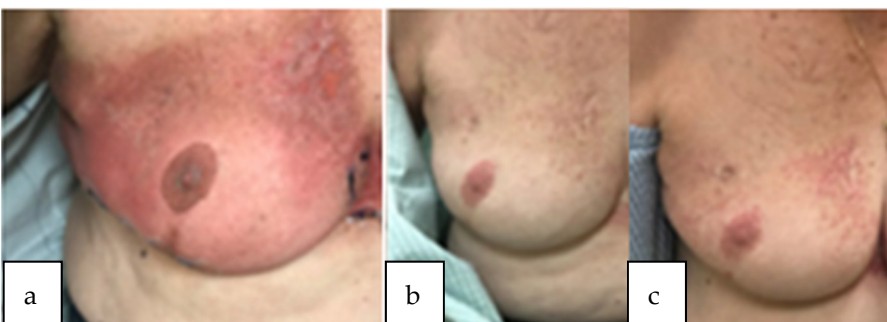

**Figure 1.** Patient with acute dermatitis grade 3 during radiation treatment (**a**) subsequently developed worsening telangiectasia at 1 year and 2 year follow-up (**b**,**c**). Course 1: mammosite balloon brachytherapy to 34 Gy and course 2: WBRT to 46.8 Gy in 26 daily fractions with 14 Gy lumpectomy cavity boost.

PD7-09 was a multi-institutional prospective study of 35 patients treated with APBI using 40.05 Gy (2.67 Gy/fraction) after second BCS in three university hospitals in Spain reported in abstract form only [35]. Inclusion criteria included age > 50 years old, tumor size < 2 cm, absence of multifocal disease, > 48 months from primary treatment and no history of major RT toxicity. At IBTR, median age was 65 years and median time to IBTR was 154 months. Tumor recurrences were DCIS in 5 patients (14.3%) and invasive carcinoma in 30 patients (85.7%). All patients were treated with wide local excision and partial-breast reRT, 67% with IMRT, 27% with 3DCRT and 6% with IORT. With a median follow-up of 37 months, there was one IBTR (2.9%) after in situ IBRT and three metastatic recurrences (8.6%), two after invasive and one after in-situ IBTR. Both metastatic progressions after invasive IBTR were hormone-receptor-positive and HER2-negative. There were two deaths not related to breast cancer. There is currently an ongoing phase II trial NCT04371913 to evaluate outcomes in patients treated with 30 Gy in 5 fractions [36].

## 4. Local Failure after Initial Partial-Breast Radiation

There is limited data on the outcomes of patients who locally relapse after APBI. Our study included 3 patients initially treated with PBI who were treated with WBI in the reRT setting. None presented with local or distant failure. In addition to the patient described above who developed grade 3 telangiectasia, the 2 other patients had no late adverse events at 1 year follow-up and patient and physician rated cosmesis were good. An Italian multicentric study evaluated management in 224 patients who presented with local failure after initial APBI with intraoperative radiotherapy with electrons [42]. A true recurrence/marginal miss (TR/MM) was defined as an IBTR within or immediately adjacent to the primary tumor site. Out of 224 patients, 50.9% (N = 114) of patients underwent mastectomy alone, 9.4% (N = 21) underwent mastectomy and radiation, 28.1% (N = 63) underwent quadrantectomy and radiation and 11.6% (N = 26) underwent quadrantectomy

alone. In the 63 patients treated with BCS and RT, re-irradiation was mainly with WBI (N = 46, median dose 45 Gy, range 28.5–60 Gy, boost in 8 cases) and, to a lesser extent, with a second APBI (N = 17). APBI consisted of intraoperative radiation therapy with electrons in 13 patients and external beam RT in the remaining 4 patients (median dose 47.5 Gy, range 37.05–50.0 Gy). Cumulative incidence of third IBTR was comparable between salvage mastectomy and repeat quadrantectomy + RT groups. In the repeat quadrantectomy + RT group, comparison between the two salvage radiation modalities showed statistically higher third IBTR rate for APBI compared with WBI (three events versus one event, the latter occurring after 10 years of follow-up) but no difference in terms of DFS, OS, or distant metastases incidence. An Italian prospective pilot study evaluated the technical and dosimetric feasibility of modified WBI which excludes the volume of the breast which had previously received intraoperative radiotherapy with electrons [43]. Nine patients had repeated quadrantectomy followed by modified WBI to 45 Gy in 20 fractions. At a median follow-up of 26 months, no patients had a local recurrence and there were no grade ≥ 3 events. Given the increasing use of APBI with increasingly expanding selection criteria, the optimal management of relapse after APBI requires further investigation.

## 5. APBI Re-Irradiation at NYU—CT Simulation, Contours, Beams, Planning Constraints and Imaging

### 5.1. CT Simulation

Patient positioning for reRT depends on treatment position (supine vs. prone) of the first course of radiation, on the reRT target volume whether whole breast or APBI and on the ability to tolerate the prone position. One consideration is the ability to register the images from each treatment if the patient is simulated and treated in the same position.

### 5.2. Target Volumes and Normal Structures

For all cases, the physician contours the tumor bed (TB), which includes the resection cavity and surgical clips (when applicable).

For APBI treatment, the planning target volume (PTV) is defined as TB with 1.5 cm 3D expansion. From the PTV a planning target volume evaluation (PTV_Eval) is created, limited to be within the defined ipsilateral breast tissue, excluding 0.5 cm of tissue under the skin (if TB is not within this region) and tissue beyond the chest wall, pectoralis muscles and lung. Additionally, the physician contours ipsilateral and contralateral breast.

For whole-breast treatments, a planning target volume tangents (PTV_Tangents) is created as the volume contained within the tangent fields (obtained from the 50% isodose line) and cropped 0.6 cm, with no overlap of heart or lung. In addition, planning target volume tumor bed evaluation (PTV_TB_Eval) is created by adding 1 cm 3D expansion around TB and cropped to be within PTV_Tangents (if TB does not extend superficially beyond PTV_Tangent volume).

Normal structures and organs-at-risk (OARs), including the ipsilateral and contralateral lung, heart and skin, are contoured by a dosimetrist and reviewed by a physician.

### 5.3. Beams

All treatment plans are generated in the Eclipse planning system (Varian Medical Systems, Palo Alto, USA) by a dosimetrist and reviewed by a physicist and physician.

For APBI, 6MV opposed photon tangents (3D-CRT or IMRT) are primarily used. However, 15 MV 3D-CRT photons are encouraged for fields in which the PTV_TB_Eval is not within the build up region.

For whole-breast treatment, we generally use 6MV opposed-photon tangents (3D-CRT or IMRT). However, for cases with large separations, a hybrid of 6MV 3D-CRT or IMRT and 15MV 3D-CRT (67%/33% dose contribution per tangent, respectively) may be used to meet the maximum dose constraint. Fifteen MV beams are not used for fields in which the tumor bed lies in the buildup region.

*5.4. Constraints*

For APBI treatments with 30 Gy in 5 fractions, the ipsilateral breast is constrained to V50% (V15 Gy) < 50–60% and V100% (V30 Gy) < 35%. Other constraints include heart V5% < 5%, ipsilateral lung V30% < 15%, contralateral lung V5% < 15%, PTV_Eval D95% > 100% and D99.5% > 90%, Tumor Bed D98% > 100%, and Body D0.03cc < 110%. For whole-breast treatments, the PTV_Tangents maximum dose is constrained to D0.03cc ≤108–112%, keeping the dose coverage to D95% ≥ 95–100%. Table 4 summarizes the dose constraints for APBI and whole-breast planning techniques.

**Table 4.** Dose objectives for APBI and whole-breast treatment planning techniques.

| Structure | APBI Dose Objectives | Whole-Breast Dose Objectives |
|---|---|---|
| Tumor Bed | D98% > 100% | D98% > 100% |
| PTV_TB_Eval | D95% > 100% D99.5% > 90% | D95% ≥ 95–100% |
| PTV_Tangents | N/A | D95% ≥ 95–100% D0.03cc ≤ 108–112% |
| Body | D0.03cc < 110% | N/A |
| Breast_Ipsilateral | V50% (V15Gy) < 50–60% V100% (V30 Gy) < 35% | N/A |
| Heart | V5% < 5% | Right Breast: D0.03cc ≤ 2000 cGy Mean ≤ 100–400 cGy Left Breast: V25Gy–V20Gy ≤ 5% Mean ≤ 200–400 cGy V20Gy ≤ 15–20% V10Gy ≤ 35–40% V5Gy ≤ 50–55% |
| Lung_Ipsilateral | V30% < 15% | |
| Lung_Contralateral | V5% < 15% | N/A |

## 6. Future Directions

Given the variability in radiation technique, dose, and fractionation, eligibility criteria for repeat BCS followed by reRT remains to be established. Most US trials included all patients ≥ 18 years old, tumor size ≤ 3 cm and a minimum one-year interval after initial BCS and RT. European trials have more stringent eligibility criteria, including age ≥ 50 years old, tumor size ≤2 cm and a minimum 4-year interval after initial BCS and RT. An Italian review of retrospective studies and the DEGRO expert panel in Germany restricted a second breast conservative approach to patients age ≥ 50 years old, tumor size ≤ 2 cm and a minimum 4-year interval after initial BCS and RT [41,42]. Given the safety and feasibility of repeat BCS followed by reRT in patients with early stage breast recurrences, a randomized controlled trial comparing the RTOG 1014 regimen (45 Gy in 1.5 Gy twice daily) with shorter course regimens (40 Gy/15 fx daily or 30 Gy/5 fx every other day) would help further define clinical suitability criteria for these hypofractionated regimens.

Furthermore, there is no standard approach for differentiating a true recurrence that persisted after definitive treatment from a new primary that arose de novo in the same or different quadrant of the breast. Studies have used concordance of tumor location, time to occurrence, hormone receptor status or tumor grade to distinguish a new primary from a recurrence with varying degrees of success [44–47]. Laird et al. showed that when classifying a new primary by the presence of an in situ component and a true recurrence by its absence, NPs had significantly better outcomes than TR, with 5-year DFS of 43% and 80% for TR and NP, respectively [48]. Lips et al. performed genomic analyses on the initial DCIS lesion and paired invasive recurrent tumors in 95 patients and found that in 75% of cases, the invasive recurrence was clonally related to the initial DCIS, suggesting that the initial DCIS tumor cells persisted after initial treatment [49]. Further research is needed

to better classify TR vs. NP in order to guide our treatment of the both the primary and recurrent disease.

**Author Contributions:** Conceptualization: C.H.A., J.I.C. and N.K.G.; writing—original draft preparation, review and editing: C.H.A., J.M.P., P.G., A.M., C.H., J.I.C. and N.K.G. All authors have read and agreed to the published version of the manuscript.

**Funding:** This research received no external funding.

**Conflicts of Interest:** The authors declare no conflict of interest.

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
