# Peer review of "Different Re-Irradiation Techniques after Breast-Conserving Surgery for Recurrent or New Primary Breast Cancer"

_curroncol, doi:10.3390/curroncol30010088_

Round 1

Reviewer 1 Report

This is a very useful overview about re-irradiation with different techniques.

This is an "hot topic" and a lot of Centres are using external beam radiotherapy, but leterature is lacking.

I think the review is well conducted and i would suggest to the authors adding this reference "The POLO (Partially Omitted Lobe) approach to safely treat in-breast recurrence after intraoperative radiotherapy with electrons"(with argumentation) in the section about "local failure after initial partial breast radiation".

thank you

Reviewer 2 Report

see attached

Round 2

Reviewer 2 Report

The manuscript is greatly improved and should be published. It is much clearer and now can be read by the interested reader that is is not a breast surgeon or radiologist.